# HELP ME EXPLORE: COMBINING AUTOTELIC AND SOCIAL LEARNING VIA ACTIVE GOAL QUERIES

## ABSTRACT

Most approaches to open-ended skill learning train a single agent in a purely sensorimotor world. But because no human child learns everything on their own, we argue that *sociality* will be a key component of open-ended learning systems. This paper enables learning agents to blend individual and socially-guided skill learning through a new interaction protocol named *Help Me Explore* (HME). In *social episodes* triggered at the agent's demand, a social partner suggests a goal at the frontier of the agent's capabilities and, when the goal is reached, follows up with a new adjacent goal just beyond. In *individual episodes*, the agent practices skills autonomously by pursuing goals it has already discovered through either its own experience or social suggestions. The idea of augmenting an individual goal exploration with social goal suggestions is simple, general and powerful. We demonstrate its efficiency on hard exploration problems: continuous mazes and a 5-block robotic manipulation task. With minimal social interventions, the HME agent outperforms both the purely social and purely individual agents.

## 1 INTRODUCTION

Open-ended learning is an important challenge in artificial intelligence (AI) where the goal is to design embodied artificial agents able to grow diverse repertoires of skills across their lives (Doncieux et al., 2018; Stooke et al., 2021). Goal-conditioned reinforcement learning (GC-RL), because it offers the possibility to train an agent on multiple goals in parallel, recently emerged as a key component in this quest (Schaul et al., 2015; Andrychowicz et al., 2017; Liu et al., 2022). But where do goals come from? Almost always, they are sampled from a fixed distribution over a predefined goal space; i.e. they come from an engineer. Beyond the heavy engineering burden it presupposes, this approach is fundamentally limited in realistic environments with infinite possibilities because engineers cannot foresee how the agents will learn or what they will be able to achieve. Instead, we must draw inspiration from the study of human learning and pursue a *developmental approach*: agents should be intrinsically motivated to learn to represent, generate, pursue and master their own goals — i.e. they must be *autotelic* Steels (2004); Colas et al. (2022).

In a recent paper introducing the autotelic framework, Colas et al. identify two challenges: 1) learning goal representations; 2) exploring the corresponding goal space and mastering the associated skills (Colas et al., 2022). Although, eventually, all autotelic agents must learn their own goal representations (challenge 1, Eysenbach et al. (2018); Nair et al. (2018b); Colas et al. (2020)), most existing approaches assume a pre-defined goal space and focus on its exploration and mastery (challenge 2, Schaul et al. (2015); Colas et al. (2019); Akakzia et al. (2020)). In this second challenge — the one we focus on — agents must learn to organize their own learning trajectories by prioritizing goals with the objective of maximizing long-term skill mastery.

Exploring and developing skills in unknown goal spaces can be hard when they are not *uniform*: some goals might be easy, others hard; most of them might be unreachable. So how can an agent select which goals to practice now? Because they direct the agent's behavior, the selected goals impact the discovery of future goals (exploration), and the mastery of known ones (exploitation) — yet another instance of the *exploration–exploitation dilemma* (Thrun, 1992). Existing methods belong to the family of *automatic curriculum learning strategies* (Portelas et al., 2020). They propose to replace the hard-to-optimize distal objective of *general skill mastery* with proxies such as intermediate difficulty (Florensa et al., 2018), learning progress (Colas et al., 2019; Blaes et al., 2019; Akakzia

et al., 2020) or novelty (Pong et al., 2019; Ecoffet et al., 2019). But all ACL strategies are so far limited to generate goals from the distribution of effects the agents already experienced. Can agents demonstrate efficient exploratory capacities if they only target goals they already achieved (Campos et al., 2020)? To solve this problem, we must once again draw inspiration from the study of human learning, this time from socio-cultural psychology.

Philosophers, psychologists, linguists and roboticists alike have argued for the central importance of rich socio-cultural environments in human development (Wood et al., 1976; Vygotsky & Cole, 1978; Berk, 1994; Tomasello, 1999; Lindblom & Ziemke, 2003; Lupyan, 2012; Colas, 2021). Humans are social beings wired to interact with and learn from others (Vygotsky, 1978; Tomasello, 1999; 2019). When they explore, it is often through socially-guided interactions (e.g. by having a parent organize the playground) and it often relies on the inter-generational population-based exploration of others, a phenomenon known as the *cultural ratchet* (Tomasello, 1999). In *guided-play* interactions (Weisberg et al., 2013; Yu et al., 2018) caretakers scaffold the learning environments of children and help them practice new skills *just beyond their current capacities*, i.e. in Vygotsky's *zone of proximal development* (Vygotsky, 1978). Can AI learn from these insights? We believe so.

This paper introduces a novel social interaction protocol for autotelic lagents named **H**elp **M**e **E**xplore (HME). This consists of a succession of *individual* and *social* episodes. In individual episodes, the agent pursues self-generated goals sampled from the set of outcomes the agent already experienced during training. In social episodes, a social partner suggests a novel goal to the agent and decomposes it into two consecutive sub-goals: 1) a *frontier goal* that the agent already discovered and, if it is reached, 2) a *beyond goal* never achieved by the agent but *just beyond the its current abilities*. The frontier goal acts as a stepping stone facilitating the discovery and mastery of the beyond goal. Combined with a powerful autotelic RL algorithm, HME makes significant steps towards *teachable autotelic agents*: autonomous agents that leverage external social signals and interweave them with their self-organized skill learning processes to gain autonomy, creativity and skill mastery (Sigaud et al., 2021). Our contributions are twofold:

- **Help Me Explore** (HME). This social interaction protocol lets agents blend social and individual goal explorations through social goal suggestions. It is generally applicable to any multi-goal environment and it is tested in a 5-block robotic manipulation domain, where it enables agents to master 5-block stacking, a notoriously hard exploration challenge.

- **A socially-sensitive autotelic agent** (HME **agent**). We augment an existing autotelic RL agent with two social capabilities: 1) an active learning mechanism allowing the agent to self-monitor its learning progress and, when it stagnates, query the social partner for a goal suggestion and 2) the ability to internalize social goal suggestions and rehearse them autonomously during individual episodes. This ability is inspired from Vygotsky's concept of internalization describing how learners model social processes occurring in the zone of proximal development to augment their future autonomy and capacities (Vygotsky, 1978). These two mechanisms drastically reduce the amount of social interactions required to achieve skill mastery.

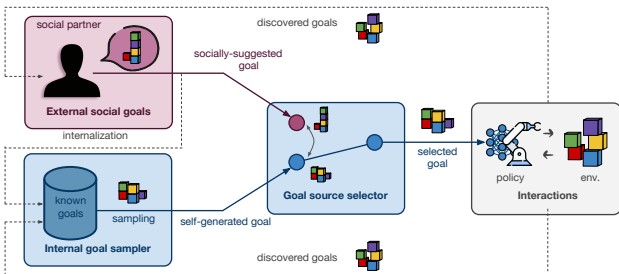

Figure 1: The HME social interaction protocol and the social autotelic agent. The autotelic agent (blue) interacts with its world and discovers new configurations (grey box, right). Through its goal source selector (blue box, middle), the agent decides whether to sample from its known goals (blue box, bottom left) or to query the social partner for a social goal (pink box, top left). Socially-suggested goals are further internalized within the agent for later use during individual episodes.

We evaluate the social autotelic agent and the HME interaction protocol in a complex block manipulation environment based on the Fetch robotic arm (Plappert et al., 2018). Whereas state-of-the-art ACL methods fail to discover difficult configurations, our agent successfully discovers and masters tens of thousands of configurations, including difficult stacks of 5 blocks and complex pyramidal structures. Thanks to its internalization and active query mechanisms, the agent requires social goal suggestions in only 0.5% of the training episodes.

## 2 RELATED WORK

**Hard exploration problems.** Environments can be said to have a *hard exploration problem* when their reward function is sparse and/or deceptive (Ten et al., 2022). Fighting hard exploration problems requires exploratory mechanisms. Indeed, simple undirected exploration such as adding random noise on the agent's actions (Mnih et al., 2015) or policy parameters (Plappert et al., 2017) is not enough to experience informative rewards in hard exploration problems. Directed exploration mechanisms involve the use of *intrinsic motivations*: auxiliary reward functions driving the agent to explore (Oudeyer & Kaplan, 2007; Ten et al., 2022). While *knowledge-based* intrinsic motivations aim at improving the agent's model of the world (Barto, 2013; Bellemare et al., 2016; Achiam & Sastry, 2017; Pathak et al., 2017; Burda et al., 2018), *competence-based* intrinsic motivations let agents target their own goals and pursue them as a means of exploration (Colas et al., 2022). Finally, the recent *Go-Explore* approach recommends to first achieve a known goal that has been sparsely visited, then to start exploring from there (Ecoffet et al., 2019; Pitis et al., 2020). This paper proposes to solve hard exploration problems with competence-based intrinsic motivations. Besides, we argue that sparsely visited areas of the goal space considered in approaches like *Go-Explore* do not necessarily represent stepping stones for new goals, and consider an expert SP that proposes these intermediate checkpoints.

**Autotelic reinforcement learning.** Autotelic agents are autonomous agents intrinsically motivated to learn to represent, generate, pursue and master their own goals (competence-based intrinsic motivations Colas et al. (2022)). More than explorers, autotelic agents are designed to build open-ended repertoires of skills across their lives. This quest first emerged in the field of developmental robotics (Steels, 2004; Oudeyer & Kaplan, 2007; Nguyen & Oudeyer, 2012; 2014b; Baranes & Oudeyer, 2013; Forestier & Oudeyer, 2016; Forestier et al., 2017) and recently converged with state-of-the-art deep reinforcement learning (RL) research (Eysenbach et al., 2018; Warde-Farley et al., 2018; Nair et al., 2018b; 2020; Pong et al., 2019; Colas et al., 2019; 2020; 2022). Following the Piagetian tradition in psychology, most of these approaches consider individual agents interacting with their physical environment and leveraging forms of intrinsic motivations to power their exploration and learning progress (Warde-Farley et al., 2018; Nair et al., 2018b; Pong et al., 2019; Colas et al., 2019; Stooke et al., 2021). A fundamental limitation of these approaches is their incapacity to generate goals outside of the distribution of known outcomes, which drastically limits their exploratory capabilities. The Imagine approach proposes to leverage linguistic structures to imagine new goals out of the distribution of known outcomes but relies on a hard-coded mechanism which does not allow the generation of truly creative goals (Colas et al., 2020). In this paper, we leverage social interactions with a more knowledgeable partner to overcome this limitation.

**Interactive reinforcement learning.** Interactive RL lets non-expert users communicate, teach and guide the learning processes of artificial agents (Thomaz & Breazeal, 2008; Argall et al., 2009; Celemin & Ruiz-del Solar, 2015; Najar et al., 2016; Christiano et al., 2017a). These methods often limit the agent's autonomy because they heavily rely on the instructor to guide the whole learning process, either through demonstrations (Nair et al., 2018a; Fournier et al., 2019; Hester et al., 2018), preferences (Christiano et al., 2017b) or guidance (Grizou et al., 2013).

**Social learning and intrinsic motivations.** Combining social learning and intrinsic motivations has been studied as a way to enhance skill acquisition in artificial agents (Schaal et al., 2003; Thomaz et al., 2006; Thomaz & Breazeal, 2008; Peters & Schaal, 2008; Kober & Peters, 2011; Stulp & Schaal, 2011; Nguyen & Oudeyer, 2014a). The main motivation is that social guidance provided by humans can drive the learner to new regions of its state space, where its own random exploration alone is usually insufficient. Intrinsic motivations can then be used to further explore these new areas. Such a combination profits from both the broader aspect of intrinsic motivations (explore as

many goals as possible) and the specialized aspect of social learning (follow a specific instruction). This was studied not only in the context of learning one single skill (Schaal et al., 2003; Peters & Schaal, 2008; Kober & Peters, 2011; Stulp & Schaal, 2011), but also in achieving a variety of goals (Thomaz et al., 2006; Thomaz & Breazeal, 2008; Nguyen & Oudeyer, 2014a). Most of these works rely on human demonstrations or upstream feedback as a form of social guidance. In this paper, social signals take the form of goals suggested by the SP at the demand of the learning autotelic agent.

## 3 METHODS

This paper studies how a social autotelic agent can efficiently explore a complex environment while relying on a minimal number of interventions from a social partner. We first describe the problem of exploring an unknown goal space (Section 3.1), before introducing the proposed social interaction protocol (Section 3.2, Figure 1) and our socially-sensitive autotelic agent (Section 3.3). The pseudo-code and implementation details can be found in Appendices A.5 and A.6 respectively.

### 3.1 PROBLEM STATEMENT

The agent evolves in a reward-free interactive world that it can perceive through *states* and act on through *actions*. This environment is modeled by an MDP: $\{\mathcal{S}, \mathcal{A}, \mathcal{T}, \rho_0\}$ where $\mathcal{S}$ is the state space, $\mathcal{A}$ the action space, $\mathcal{T} : \mathcal{S} \times \mathcal{A} \to \mathcal{S}$ the transition function describing the world's physics and $\rho_0$ the distribution of initial states. In the autotelic framework, the goal space and associated reward functions are not specific to the environment but represented within the agent; which is why they are not modeled in the MDP (Colas et al., 2022).

In this paper, the agent is given an extra representation function mapping states to high-level discrete configurations $\phi : \mathcal{S} \to \mathcal{C}$ with $\mathcal{C}$ the set of reachable configurations. Goals are then defined as target configurations the agent attempts to reach, such that the goal space $\mathcal{G}$ is equal to $\mathcal{C}$. The agent does not know $\mathcal{G}$, but can only sample goals from the set of encountered configurations $\mathcal{G}_{\text{known}} \subset \mathcal{G}$. Goal-specific reward functions are then defined as binary truth functions assessing the equality between the agent's current configuration $c(s)$ and its goal $g$: $r(g, s) = c(s) == g$ where $s$ is the current state.

We define the *exploration problem* as the discovery and mastery of a maximum number of goals within the configuration space. We do not assume access to the set of reachable configurations $\mathcal{G}$. There might be too many and the agents could surprise us with their learning abilities. In this case, we can only use a *subjective* evaluation metric. We propose to measure skill mastery on a pre-defined subset of goals we, as experimenters, find interesting. In the Experiments, Section 4.1 will introduce the multi-object manipulation environment we consider, its associated goal space and the specific evaluation metrics.

### 3.2 THE SOCIAL INTERACTION PROTOCOL

The *Help Me Explore* interaction protocol (HME) is centered around the idea of social goal proposal. It involves three elements: an environment, an autotelic agent, and a social partner (SP), see Figure 1. The environment is a reward-free MDP as defined in the previous section. The autotelic agent is a goal-conditioned lagent that also embeds goal representations and associated reward functions (see description in Section 3.3).

The *social partner* (SP) is a hand-coded more-knowledgeable other. We assume that it shares the same goal representations as the agent and that it knows a subset of the achievable goals ($\mathcal{G}_{SP} \subset \mathcal{G}$) and how to decompose them into simpler sub-goals (e.g. a stack of two is a first step towards a stack of three). As the agent trains, the SP constructs a model of its current exploration limits by tracking which goals from $\mathcal{G}_{SP}$ the agent achieved and which it did not. To suggest a goal, it samples a target configuration lying *just beyond the agent's current abilities* (Beyond goal). It then decomposes this *Beyond goal* by suggesting a *Frontier goal*: a stepping stone on the way of the Beyond goal that the agent already achieved during training. As it does so, the SP builds a *zone of proximal development*; it helps the agent achieve goals beyond its current limits by adapting its suggestion to the agent's current knowledge. Appendix A.2 provides a detailed description of the SP code.

The HME interaction protocol starts after an initialization phase of $N_i = 10000$ episodes where the agent performs random actions to discover its first configurations and, thus, its first potential goals. In HME, the agent engages either in *individual episodes* where it pursues its own goals or in *social episodes* where it pursues a goal suggested by the SP. Although this work uses a hand-coded SP, the discussion section addresses the idea of replacing it with a human partner.

### 3.3 A SOCIALLY-SENSITIVE AUTOTELIC AGENT

The socially-sensitive autotelic agent is built on top of a powerful autotelic agent called GANGSTR presented in Akakzia & Sigaud (2022). We augment it with *social capabilities*: the ability to select between self-generated and social goals and the ability to *internalize* social goals for later reuse.

**The GANGSTR agent.** GANGSTR is a goal-conditioned reinforcement learning agent with graph neural network (GNNs) architectures trained with the soft actor-critic algorithm (Haarnoja et al., 2018). The GNNs architectures for the policy and value network let us encode relational biases which drastically increase the agent's transfer capabilities, especially in multi-object manipulation tasks (see details in Appendix A.4 and in the original paper Akakzia & Sigaud (2022)). Because goals are only rarely achieved by chance (sparse reward problem), we reuse the multi-criteria hindsight experience replay mechanism that was shown to significantly increase sample efficiency in Lanier et al. (2019); Akakzia & Sigaud (2022). We further augment the autotelic agent with two social capabilities: 1) the ability to decide when to ask the SP for a goal proposal and 2) the ability to internalize socially-suggested goals for later reuse.

**Active goal queries based on learning progress.** The agent tracks its overall learning progress (LP) towards known goals and computes the probability to query the SP for a goal as a function of LP. The general principle is that, whenever LP is low, the agent is most likely neither learning nor discovering anything new on its own and should ask for a social goal suggestion. To estimate its LP, the agent first estimates its competence. To this end, we train a neural network to estimate the success or failure of reaching a goal given the goal embedding with a binary classification loss. Every $N_e = 6000$ training episodes, the agent estimates its competence by computing the projected success rate averaged over a subset of goals $\mathcal{G}_{\mathrm{LP}}$ uniformly sampled from the set of known goals $\mathcal{G}_{\mathrm{known}}$. LP is then computed as the difference between two competence measures separated by a fixed time period $N_e$: $\mathrm{LP}^t = \mathrm{C}^t - \mathrm{C}^{t-N_e}$. The query probability $P^t_{\mathrm{query}}$ at training episode $t$ is then defined by:

$$P^t_{\mathrm{query}} = exp\Big( - \beta \sum_{g \in \mathcal{G}_{\mathrm{LP}}} \mathrm{LP}^t_g \Big), \tag{1}$$

where $\beta$ is a hyper-parameter controlling the tolerance to a lower LP. As $\beta$ grows, the agent becomes less susceptible to query the SP and will only do so for lower LP values.

**Social goal internalization.** In Vygotsky's theory, children learn to *internalize* social processes occurring in the zone of proximal development and to turn them into internal cognitive tools such that they can solve hard problems themselves in the future (Vygotsky, 1978). Our social autotelic agents model this internalization by storing the pairs of *Frontier* and *Beyond* goals suggested by the SP in their memory of known goals, even when those were not achieved in practice. This offers them the ability to resample and practice these goals on their own during individual episodes which lowers the number of social interactions required from the SP.

**Training organization.** At each new episode, the agent estimates its current learning progress and derives the corresponding probability to query the SP. Sampling from this probability, the agent decides whether to pursue a self-generated goal (individual episode) or to query the SP for a goal suggestion (social episode). Self-generated goals are sampled uniformly from the set of known goals $\mathcal{G}_{\mathrm{known}}$ which includes the set of goals internalized from social suggestions. The agent then pursues the goal and computes its own rewards by comparing its reached configurations with the goal configuration. After adding the set of collected transitions to a replay buffer, we run the soft actor-critic algorithm (Haarnoja et al., 2018) augmented with the multi-criteria hindsight experience replay mechanism (Lanier et al., 2019) to update the policy and value networks.

## 4 EXPERIMENTS

This experimental section demonstrates the benefits of blending social and individual exploratory processes with respect to purely social or purely individual explorations, including those powered by state-of-the-art ACL strategies. We first conducted preliminary experiments in simple 2D continuous mazes involving exploration problems (see details and experiments in Appendix A.1). These experiments prove the benefits of sociality for skill learning, which led us to tackle another hard exploration problem in a complex robotic manipulation setting. We open-source an anonymized version of our code via Anonymous Github.

### 4.1 A HARD EXPLORATION PROBLEM

We consider the *Fetch Manipulate* environment where the agent is embodied in a 4-DoF robotic arm facing 5 colored blocks on a table (Akakzia & Sigaud, 2022). The agent controls the Cartesian velocity of its gripper and gripper closure (4 dimensions) and perceives the position and velocity of the blocks and gripper. In addition, it perceives a 35D binary representation of the blocks configuration obtained by applying the three predicates *above* (binary non-symmetric), *close* (binary symmetric) and *on_table* (unary) on all possible pairs of blocks. The objective of the agent is to explore the resulting configuration space ($2^{35}$ configurations). All episodes start with all blocks in contact with the table (coplanar configuration).

This exploration problem is hard for several reasons. First, the goal space is incredibly large. Although it is hard to estimate the amount of reachable configurations, the following experiments will show that good agents can discover a diversity of these goals. Second, the large majority of goals cannot be reached but the agent cannot tell *a priori*. Third, some goals are much harder to reach than others (e.g. stacks of $5 >$ coplanar configurations). In fact, the hardest goals cannot be reached by chance following a random exploration because they involve time-extended behaviors (stacking 5 blocks one after the other). Up to this day, learning 4+ block stacking often requires specific mechanisms such as hand-defined curricula (Li et al., 2019), demonstrations (Nair et al., 2018a) or reward shaping (Popov et al., 2017). Without these specific mechanisms, our purely individual agent only discovers stacks of up to 4 blocks, which is the highest we found in the literature (Akakzia & Sigaud, 2022).

How can an agent efficiently explore that configuration space? Before we answer that question, we shall define our evaluation metrics. Indeed, we cannot list the achievable goals *a priori* and therefore need to define a subjective set of evaluation goals. We evaluate the quality of an agent along two dimensions:

- **Performance metrics**: average success rates computed over 10 sets of interesting evaluation goals that humans could easily describe, including coplanar configurations, stacks of 2, 3, 5, pyramids and their combinations (see the classification in Table 1). At each evaluation, the agent is given 20 goal configurations uniformly sampled from each of the 10 evaluation classes (200 goals). Here, we report both the 10 per-class success rate and the global success rate computed as their average (SR).

- **Social metric**: the amount of social interactions involved in the training of socially-sensitive agents, as a percentage of all training episodes.

| Class | $C_1$ | $C_2$ | $S_2$ | $S_3$ | $S_2 \& S_2$ | $S_2 \& S_3$ | $P_3$ | $P_3 \& S_2$ | $S_4$ | $S_5$ |
|---|---|---|---|---|---|---|---|---|---|---|
| # Goals | 10 | 45 | 20 | 60 | 60 | 120 | 30 | 60 | 120 | 120 |

Table 1: Classes of evaluation goals. $C_i$ goals regroup all coplanar configurations where exactly $i$ pairs of blocks are *close*. $S_i$ goals have exactly $i$ stacked blocks. $P_i$ goals have a pyramid made of $i$ blocks. The *&* symbol is a logical *AND*. All unspecified predicates are set to *false*. Classes are disjoint sets and their union does not cover the whole configuration space.

In all subsequent experiments, we train our agents for a fixed budget of 20 hours and periodically evaluate them offline without exploration noise. For all conditions, we report the mean and standard deviation computed over 5 seeds. Stars indicate statistical differences with respect to the first condition (leftest in the legend) using Welch's t-test with $\mathcal{H}_0$: no difference in the means and $\alpha = 0.05$.

## 4.2 DOES THE AGENT NEED A SOCIAL PARTNER?

How can we tackle this hard exploration problem? The standard autotelic approach would be to let the agent sample goals uniformly from the set of configurations it already discovered (autotelic baseline). If this does not work, one could augment the agent with an ACL strategy to refine the goal selection mechanism. Here, we consider two baselines inspired from state-of-the-art ACL algorithms: 1) the selection of goals associated with high learning progress (LP baseline, based on Colas et al. (2019)) and 2) the selection of goals the agent is the most uncertain about (value-disagreement sampling baseline (VDS), based on Zhang et al. (2020)). In the LP baseline, we train a neural network to predict the success or failure for any given goal (binary classification, see Section 3) and use it to estimate LP. After having estimated LP for a set of 50 goals sampled uniformly from $\mathcal{G}_{known}$, the agent pursues the one with the highest LP. In the VDS baseline, we train 3 Q-value networks instead of one and use their disagreement as a measure of the agent's uncertainty. After having estimated the value disagreement for 1000 goals uniformly sampled from $\mathcal{G}_{known}$, the agent samples a goal using the softmax distribution of their disagreement scores. Hyperparameters were taken from the original papers (Colas et al., 2019; Zhang et al., 2020).

Figure 2a shows that our proposed algorithm outperforms both the standard autotelic baseline and the two ACL baselines inspired by state-of-the-art algorithms (Colas et al., 2019; Zhang et al., 2020). Although all conditions seem to demonstrate similar skill learning capacities at first, non-social baselines reach a plateau around a SR of 0.75 while HME reaches a score of 0.93. As shown in the top row of Figure 3, ACL agents do not even discover the hardest configurations. They are indeed focused on the goals that are currently the most challenging for the agent, which do not represent good stepping stones for the discovery of more complex configurations.

Social interactions do not only suggest interesting stepping stones (frontier goal), they also point to the adjacent novel goal when the stepping stone is reached (beyond goal). To test the importance of the frontier goals, we introduce HME-50 B, an ablation of our method where the SP only suggests beyond goals. Figure 2b depicts the counts of stepping stones during training for the full HME-50 and its ablation. By stepping stones, we refer to configurations reached by the agent that are adjacent to unknown configurations in the structured goal space of the SP. Although both methods perform similarly in terms of success rates (orange and blue curves in Figure 2a), HME-50 explores faster than HME-50 B. This suggests that the role of the frontier goal is to enable the agent to reach the beyond goal from the first time. The *Go-Explore* approach from Ecoffet et al. (2019) leverages a similar decomposition. In *Go-Explore*, the stepping stone is selected using a *novelty proxy* and the following exploration is obtained with random actions. Could this work in our hard exploration problem?

To test this, we compared four ways of selecting stepping stones: 1) uniformly from the set of known configurations; 2) with a bias towards sparse regions of the goal space (novelty proxy); 3) with a bias towards high-LP goals (LP proxy) and 4) in the same way as HME; i.e. on the way to a beyond goal (social bias). Just like in Go-Explore, all variants first pursue a stepping stone, then explore with random actions from there. Variant 2 is the closest to the original Go-Explore implementation from Ecoffet et al. (2019). Figures 2a and 3 show that the HME agent outperforms

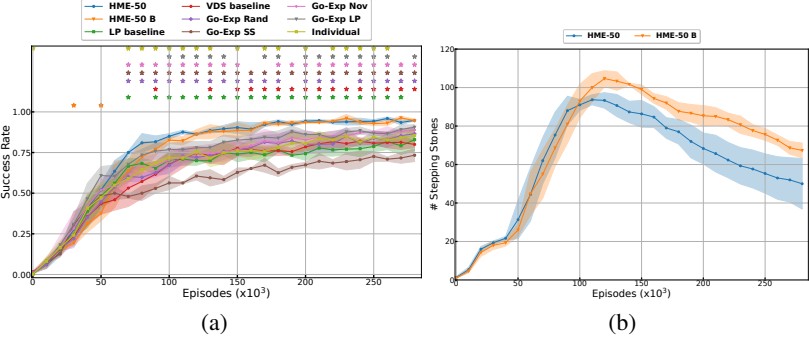

(a)              (b)

Figure 2: a) Global success rates for HME and different baselines. b) Count of stepping stones for HME-50 with and without the frontier goal suggestions.

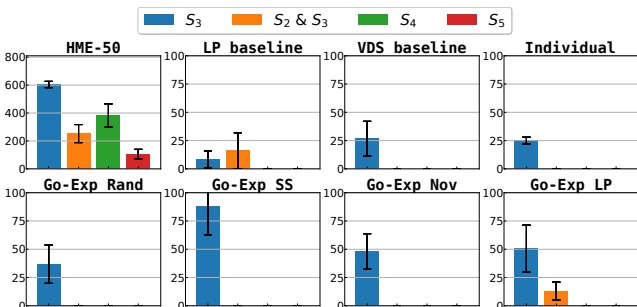

Figure 3: Exploration metrics for HME, ACL baselines and Go-Explore baselines. Counts of configurations discovered across training for the hardest classes of evaluation goals.

all these variants and discover a wider diversity of configurations. Comparing condition 4) with conditions 1-3) indicates that the social selection of stepping stone is not enough to power an efficient exploration. Interestingly, we find that HME outperforms condition 4, its variant which replaces exploration towards the beyond goal with random actions. Altogether, these results indicate that HME agents benefit from directing their exploration phase towards a socially-suggested beyond goal instead of exploring with random actions.

The results presented in this section show that ACL methods and go-explore strategies cannot replace social goal suggestions. The increased exploratory capabilities originate in the combination of returning to a known stepping stone and performing a subsequent exploration directed towards a beyond goal. Only social interactions can give agents access to such knowledge.

### 4.3 HOW MUCH SOCIAL INTERACTION DOES THE AGENT NEED?

The active query mechanism lets the agent decide when it needs a social goal suggestion from the SP. The amount of social episodes is thus a function of both the sensitivity of the agent to a stagnation in its perceived LP ($\beta$ in Equation equation 1) and of the agent's actual progress in reaching its goals. For $\beta = 0$, the agent is purely social and asks for suggestions all the time (social baseline). For $\beta = \infty$, the agent is purely individual and never asks for suggestions (individual baseline). In between, the agent is both individual and social (HME-$\beta$ for $\beta$ in $[20, 50, 100, 200]$.

Table 2: Evaluation metrics for HME variants as a function of their social sensitivity $\beta$.

| Metrics \ Agent | Social ($\beta=0$) | HME-20 | HME-50 | HME-50 w/o intern. | HME-100 | HME-200 | Indiv. ($\beta=\infty$) |
|---|---|---|---|---|---|---|---|
| % Social Ep. | 100 | $51.62 \pm 2.49$ | $\mathbf{6.98 \pm 0.58}$ | $9.12 \pm 1.10$ | $0.5 \pm 0.06$ | $0.001 \pm 0.001$ | 0 |
| Global SR | $0.81 \pm 0.01$ | $0.92 \pm 0.01$ | $\mathbf{0.93 \pm 0.02}$ | $0.89 \pm 0.01$ | $0.87 \pm 0.03$ | $0.72 \pm 0.03$ | $0.75 \pm 0.05$ |

Table 2 presents the empirical share of social episodes and the global SR for all variants of the HME algorithm and the extreme baselines (purely individual and purely social). Surprisingly, the highest global performance is recorded by HME-$\beta = 50$, an agent experiencing only 6.98% of social episodes. HME-$\beta = 100$ only requires 0.5% of social episodes to outperform both the purely social and the purely autotelic agents. Figure 4 shows the evolution of per-class SR as a function of training time. The more individual variants (right) struggle in learning the most complex goals such as the $S_4$ and $S_5$ classes. If the purely social baselines makes some progress in these classes, it remains less performant and sample-efficient than intermediate variants of HME for $\beta$ in $[20 - 100]$. Because they blend individual and social episodes, these variants manage to learn the most complex goals. The $\beta = 100$ variant, especially, makes important progress on the hardest tasks with only 0.5% of social episodes.

Figure 5 presents the distribution of configurations achieved by the agents during training. We focus on the most complex configurations that can hardly be discovered via random actions: $S_3$, $S_2 \& S_3$, $S_4$ and $S_5$. The individual baseline, the social baseline and HME-$\beta = 200$ fail to discover and achieve these goals. On the other hand, HME-$\beta = 20, 50, 100$ agents are able to reach these configurations more often. Note that HME-$\beta = 50$ reaches them often although it benefits from social episodes only 6.98% of the time.

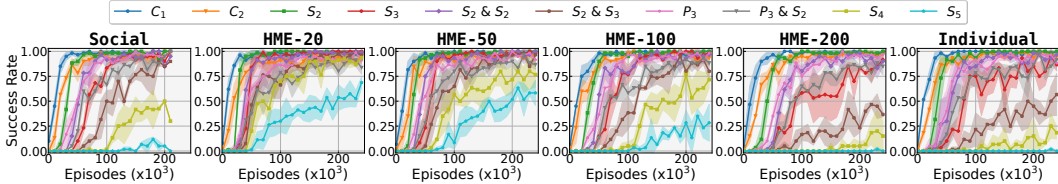

Figure 4: Per-class SR across training episodes for different social sensitivities, from high (low $\beta$) to low (high $\beta$).

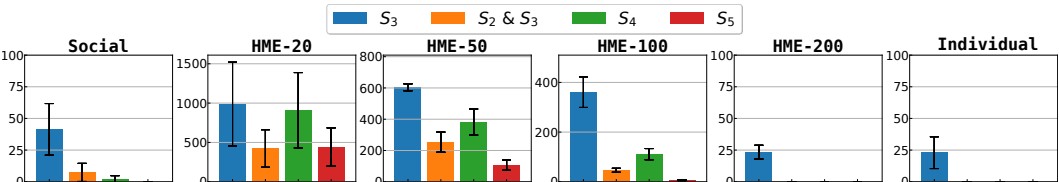

Figure 5: Exploration metrics for HME variants. Counts of configurations discovered across training for the hardest classes of evaluation goals

## 5 DISCUSSION AND CONCLUSION

This paper makes a step towards *teachable autonomous agents*: the cognitive development of artificial agents must take its root in both social situatedness and the free exploration of the physical world (Sigaud et al., 2021). It introduces two contributions: 1) an autotelic agent that learns to discover and master thousands of semantic configurations, and 2) a social interaction protocol where a social partner efficiently catalyzes the development of the learner.

The agent decides when to query the social partner, which then provides a goal just beyond the current agent's capabilities and offers an easy step on the way (frontier goal). This protocol efficiently models Vygotsky' concept of *zone of proximal development*; a specific type of interactions where a more knowledgeable other scaffolds the experience of a learner to implicitly teach them to achieve new kinds of goals and explore their world (Vygotsky, 1978). As we show, these social interactions bring significant advantages in complex worlds where exploration is hard. On the other hand, ACL methods alone are limited to sampling within the set of known goals, which drastically limits the exploration capabilities of autotelic agents.

More generally, human exploration is an inter-generational population-based process supported by social interactions and cultural evolution, a phenomenon known as the cultural ratchet (Tomasello, 1999). If we ever want to reproduce human exploration abilities in artificial agents, we need to let them interact with social partners. From them, they can learn which goals are culturally important; they can discover new goals they never reached and learn to recognize good stepping stones towards harder goals.

Although the social partner is hand-coded in this work, we hope to let real humans fill this role for our agents. Humans indeed have the natural tendency to adapt their social interactions with young learners as a function of the learner's capabilities, knowledge and goals (Vygotsky, 1978). Humans do not represent configurations with similar binary vectors in their mind, but could easily demonstrate a target configuration (beyond goal) and a stepping stone configuration that they believe the agent already knows (frontier goal). From these configurations we could extract binary target configurations to suggest to the learning agent. The principal limiting factor is the amount of required feedback. Although we drastically limit the number of required interactions in this work (0.5% already leads to significant improvements), it currently remains too high to involve actual humans (250k episodes gives 12500 social episodes). Better learning architectures with higher transfer capabilities and improved sample efficiency may help bring this number down and open the way for experiments with humans in the loop.

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

# A    APPENDIX

## A.1    PROOF OF CONCEPT EXPERIMENTS ON MAZES

In our preliminary experiments, we ran the HME agent in a set of continuous mazes that present hard exploration problems. These experiments act as a proof of concept for the more complete experiments presented in the main paper.

**Environments.**    We consider a set of three maze navigation environments generated with an existing implementation[1] based on the MuJoCo physics engine (Todorov et al. (2012), see Figure 6). The agent is represented as a moving point (orange point) that navigates specific areas (blue areas) delimited by thick walls (grey areas). Observations include the agent's position, velocity and orientation (6D). Actions control the velocity and orientation of the agent (3D). Goals include target positions for the agent to achieve (2D). Rewards are sparse such that the agent receives a reinforcing signal of +1 whenever they are within a threshold $\delta = 1cm$ from the target position.

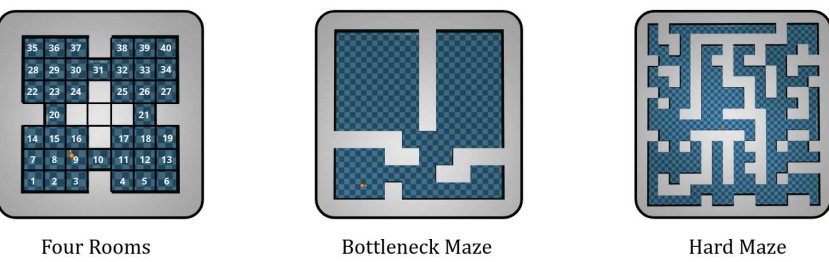

Four Rooms                    Bottleneck Maze                    Hard Maze

Figure 6: The 3 mazes considered in our study. They are of increasing difficulty in exploration.

The agent is given an extra representation function $\phi$ mapping the state to a discrete representation corresponding to the identity of an area or cell.

**Experiments.**    These experiments demonstrate the benefit of using the HME interaction protocol. We use a simpler version of the HME agent that replaces the active query mechanism with a fixed probability to query the social partner and which does not internalize socially-suggested goals. On the one hand, we use simple flat MLP networks to model the agent's internal models, including its actor and critic. On the other hand, to model the SP's suggested goals, we construct and annotate cells from the bottom row upwards (see Figure 6). The goals that the SP can propose correspond to the centers of these annotated cells. As the goal space is not known *a priori*, the agent has to 1) explore to discover goals and 2) exploit to learn to reach goals. At the beginning of each episode, the agent starts at the center of cell number 1.

We experiment with different levels of sociality: SP $p$% where p is the probability to query the social partner (SP) with $p \in \{0, 1, 10, 100\}$%. SP 0% and SP 100% represent the individual and social baseline respectively. We use three metrics to evaluate the strategies: 1) the global success rate across all the goal space (known & unknown goals), 2 the local success rates for each goal (heat maps on the mazes) and 3) the goal space coverage (ratio of discovered goals).

**Results.**    Figure 7, presents the results for the first metric (global success rate across all goal space). The results unequivocally show that the presence of the SP helps the agent reach a better performance. With only 10% of social episodes, the agent is able to master all goals on *Corridor* and *Four Rooms*. Increasing the number of social episodes does not further improve performance, as the agent with 100% and 10% social episodes performs similarly. Notably, using only 1% of social episodes is enough to show satisfying performance, further highlighting the importance of the social partner to discover new goals.

In order to illustrate that the discovery of new goals drives performance, we analyzed the exploration metrics across trained agents in Figures 8 and 9. Note that adding as little as 1% of social episodes in

---

[1]https://github.com/kngwyu/mujoco-maze

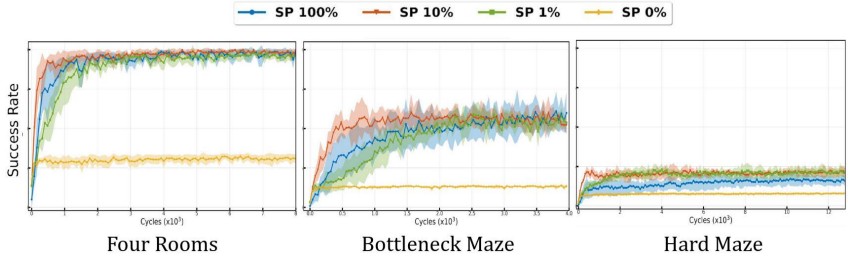

Figure 7: Global success rate across all the goal space (known & unknown goals) on the four tested mazes.

all mazes drastically improves the local success rate on goals that go beyond the first room. Indeed, thanks to the SP, the agents are able to pass the bottleneck of the maze and discover the other rooms, while SP 0% stays in the first room because it is not able to explore efficiently. The ratio of discovered goals as a function of time plotted in Figures 8 and 9 shows that the agents with social episodes rapidly discover most of the goals while the agent without social episodes does not. For the Hard maze (Figure 9, which is considered the most complex among the set of mazes we have chosen, 1% of social episodes enables the ratio of discovered goals to catch up with the ones with 10% and 100% of social episodes. However, the agent is still unable to maximize its SR on all the cells (middle part of Figure 9).

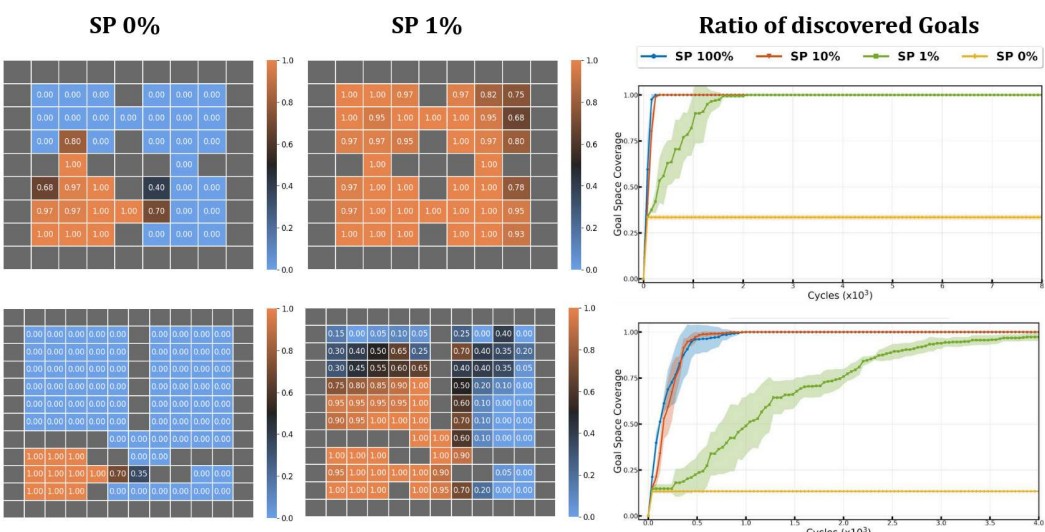

Figure 8: Performance Metrics on the Four Rooms (first row) and Bottleneck mazes (second row). Left: SR per cell with 0% of social interventions; Middle: SR per cell with 1% of social interventions; Right: Coverage of the goal space computed as the ratio of discovered goals.

## A.2 REPRESENTING THE SOCIAL PARTNER'S KNOWLEDGE

We model the social partner (SP) by a hard-coded program endowed with the set of goals that are potentially achievable by agent within the five-object manipulation benchmark. We choose to represent this domain knowledge as a directed semantic graph. This facilitates determining stepping stones in the agent's capabilities. However, it is impractical to manually enlist all the imaginable configurations and decide whether a pair should be linked or not in a graph. Semantic configurations can be infeasible for two reasons: 1) they are semantically impossible to obtain—e.g. two objects cannot both be above each other 2) they are physically impossible to achieve—e.g. in the case of inverted pyramids.

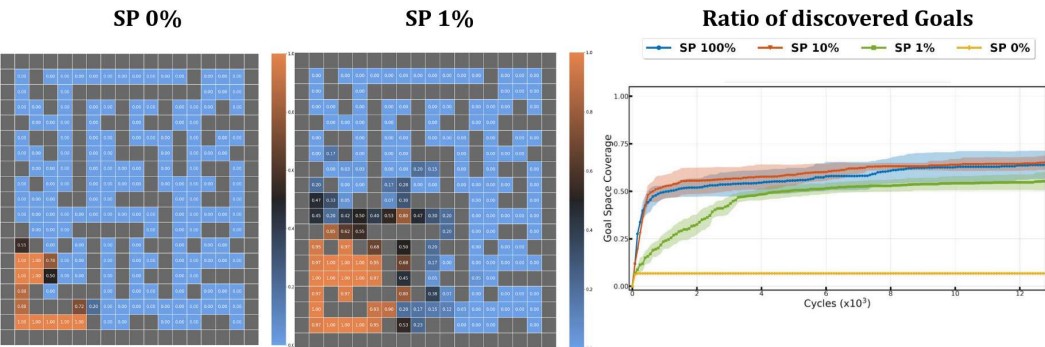

Figure 9: Performance Metrics on the Hard maze. Left: SR per cell with 0% of social interventions; Middle: SR per cell with 1% of social interventions; Right: Coverage of the goal space computed as the ratio of discovered goals.

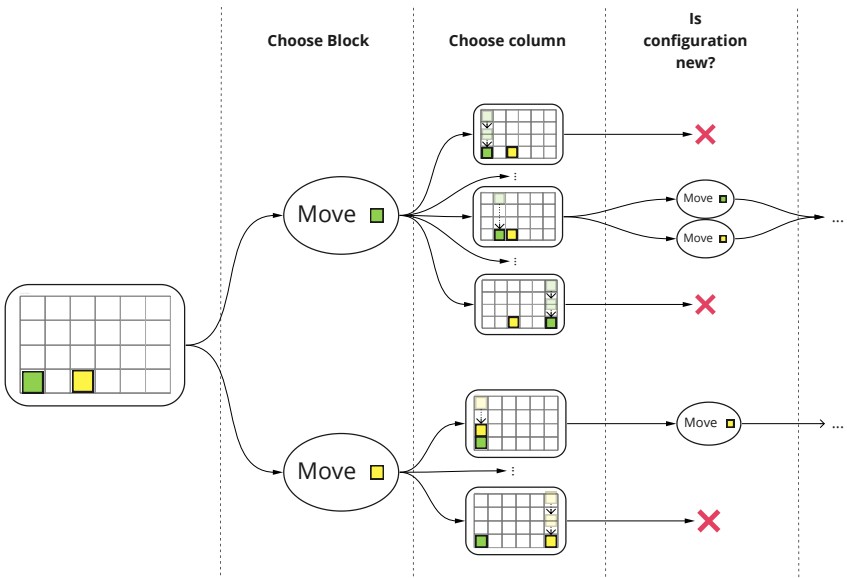

Figure 10: Simplified example with 2 blocks in a 2D grid of the oracle tree construction procedure

To avoid enlisting all semantic and physical constraints, we define two types of the SP's goal space:

- **Oracle Goal Space**: we design a 3D grid within which each cell can contain one object. Initially, all the blocks are initialized in different columns of the grid so that they are all far from each other (*root* node). From this state, we can select one object and move it to another column. If that column already contains another object, than the first one will be stacked above the second. By doing one action at each step, we can extract the current semantic configuration and link it to the previous one in the oracle graph. Iteratively repeating this process yields a tree starting from the *root* node. See Figure 10 for a 2D illustration of the described process for two blocks. The number of nodes within the oracle goal space is valued to 12666.

- **Minimal Goal Space**: this represents a sub-graph of the oracle goal space defined above. It consists of a set of goal configurations exclusively including the path from no stacks at all to stacks of 5 objects for each permutation of objects (No stack at all → Stack of 2 → ... → Stack of 5). The total number of nodes within this minimal goal space is valued to 321.

In our main experiments, we use the *minimal goal space* to represent the SP's knowledge in all our agents except for the *social baseline*. In fact, our experiments have shown that the oracle goal space

is needed when training includes *exclusively* social episodes. Note that this engineered process only serves to evaluate the capacities of the agent and is not used by the agent itself in any way.

## A.3 EVALUATION CLASSES

Table 3: The different semantic classes used in evaluation. The class *Close i* regroups all semantic configurations where $i$ pairs of blocks are close.

Table 3 illustrates the 11 evaluation classes presented in the main paper. For the sake of simplicity here, we only represent the blocks that are concerned by the underlying predicates. All the predicates associated with the other blocks have values set to 0. We use a hard-coded function to generate a random configuration given the identifier of the considered class. We also use a dictionary where keys are configurations and values are identifiers of the classes to keep counts of either the SP proposed goals or the agent's encountered and achieved goals.

## A.4 OBJECT-CENTERED ARCHITECTURE IN FETCH MANIPULATE

In the *Fetch Manipulate* environment with 5 blocks we use the GANGSTR agent proposed in Akakzia & Sigaud (2022). It perceives:

1. the low-level geometric states. Since the 5 objects share the same attributes dimensions (positions, velocities, orientations), the behavior with respect to an object should be independent from the object's attributes.
2. the high-level semantic configurations. Since the relations between all the pairs of objects share the same predicates (*close*, *above*), the behavior with respect to a binary semantic relation should be independent from the considered pair.

Thus, it is natural to encode both object-centered and relational inductive biases in our architecture. To do so, we model both the policies and critics of the agents as message passing graph neural networks (MPGNNs) (Gilmer et al., 2017). We consider a graph of 5 nodes, each representing a single object. All the nodes are interconnected, yielding a compact graph of 20 directed edges. Furthermore, we consider the agent's body attributes as global features of the policy networks and both the agent's body attributes and the actions as global features of the critic networks. A single forward pass through this graph consists in three steps:

1. **Message computation** is performed for each edge. The features of the considered edge are concatenated with the features of the edge's source and target nodes before being fed to a first shared neural network $NN_{\text{edge}}$.
2. **Node-wise aggregation** is performed for each node. The features of the considered node are concatenated with an aggregation of the updated features of all the incoming edges. The resulting vectors are then concatenated with the global features of the graph before being fed to a second shared neural network $NN_{\text{node}}$.
3. **Graph-wise aggregation** is performed once for all the graph. The updated features of all the nodes of the graph are aggregated and fed to a third neural network $NN_{\text{readout}}$.

The aggregating function needs to be permutation-invariant. We use max pooling for the *node-wise aggregation* and summation for the *graph-wised aggregation*. The final output of $NN_{\text{readout}}$ is either the action (in the case of the actor) or the $Q$-value (in the case of the critic).

## A.5 PSEUDO CODE

Algorithms 1 and 2 present the high-level pseudo-code for the individual and social learning episodes.

| **Algorithm 1** Individual Learning |
|---|
| 1: **Require** Env $E$, |
| 2: Initialize policy $\Pi$, semantic graph $\mathcal{G}_s$, path estimator $PE$, buffer $B$. |
| 3: **loop** |
| 4:     $g \leftarrow \mathcal{G}_s$.sample_node() |
| 5:     $path \leftarrow PE$.sample_path$(g, \mathcal{G}_s)$ |
| 6:     **loop** $g_i \in path$ |
| 7:         $trajectory \leftarrow E$.rollout$(g_i)$ |
| 8:         $\mathcal{G}_s$.update$(trajectory)$ |
| 9:         $PE$.update$(trajectory)$ |
| 10:        $B$.update$(trajectory)$ |
| 11:     $\Pi$.update$(B)$ |
| 12: **return** $\Pi, PE, \mathcal{G}_s$ |
| 13: |
| 14: |
| 15: |
| 16: |
| 17: |
| 18: |
| 19: |
| 20: |

| **Algorithm 2** Social Learning |
|---|
| 1: **Require** Env $E$, social partner $SP$ |
| 2: Initialize policy $\Pi$, semantic graph $\mathcal{G}_s$, path estimator $PE$, buffer $B$. |
| 3: **loop** |
| 4:     $g \leftarrow SP$.propose_goal$(\mathcal{G}_s)$ |
| 5:     $path \leftarrow PE$.sample_path$(g, \mathcal{G}_s)$ |
| 6:     **loop** $g_i \in path$ |
| 7:         $trajectory \leftarrow E$.rollout$(g_i)$ |
| 8:         $\mathcal{G}_s$.update$(trajectory)$ |
| 9:         $PE$.update$(trajectory)$ |
| 10:        $B$.update$(trajectory)$ |
| 11:     **if** $g$ is a *stepping stone* and is reached **then** |
| 12:         $g_b \leftarrow SP$.propose_unknown$(\mathcal{G}_s, g)$ |
| 13:         $path \leftarrow PE$.sample_path$(g_b, \mathcal{G}_s)$ |
| 14:         **loop** $g_i \in path$ |
| 15:            $trajectory \leftarrow E$.rollout$(g_i)$ |
| 16:            $\mathcal{G}_s$.update$(trajectory)$ |
| 17:            $PE$.update$(trajectory)$ |
| 18:            $B$.update$(trajectory)$ |
| 19:     $\Pi$.update$(B)$ |
| 20: **return** $\Pi, PE, \mathcal{G}_s$ |

## A.6 IMPLEMENTATION DETAILS

This part includes details necessary to reproduce the results. An anonymous version of our code will be made available via Anonymous Github.

GNN-*based networks.* Our object-centered architecture uses two shared networks, $NN_{\text{edge}}$ and $NN_{\text{node}}$, respectively for the message computation and node-wise aggregation. Both are 1-hidden-layer networks of hidden size 256. Taking the output dimension to be equal to $3\times$ the input dimension for the shared networks showed better results. All networks use ReLU activations and the Xavier initialization. We use Adam optimizers, with learning rates $10^{-3}$. The list of hyperparameters is provided in Table 4.

*Parallel implementation of* SAC-HER. Our experiments rely on a *Message Passing Interface* (Dalcin et al., 2011) to exploit multiple processors. Each of the 24 parallel workers maintains its own replay buffer of size $10^6$ and performs its own updates. Updates are summed over the 24 actors and the updated actor and critic networks are broadcast to all workers. Each worker alternates between 10 episodes of data collection and 30 updates with batch size 256. To form an epoch, this cycle is repeated 50 times and followed by the offline evaluation of the agent.

Table 4: Hyperparameters used in GANGSTR.

| Hyperparam. | Description | Values. |
|---|---|---|
| $nb\_mpis$ | Number of workers | 24 |
| $nb\_cycles$ | Number of repeated cycles per epoch | 50 |
| $nb\_rollouts\_per\_mpi$ | Number of rollouts per worker | 10 |
| $rollouts\_length$ | Number of episode steps per rollout | 40 |
| $nb\_updates$ | Number of updates per cycle | 30 |
| $replay\_strategy$ | HER replay strategy | $final$ |
| $k\_replay$ | Ratio of HER data to data from normal experience | 4 |
| $batch\_size$ | Size of the batch during updates | 256 |
| $\gamma$ | Discount factor to model uncertainty about future decisions | 0.99 |
| $\tau$ | Polyak coefficient for target critics smoothing | 0.95 |
| $lr\_actor$ | Actor learning rate | $10^{-3}$ |
| $lr\_critic$ | Critic learning rate | $10^{-3}$ |
| $\alpha$ | Entropy coefficient used in SAC | 0.2 |
| $\alpha_{EMA}$ | EMA coefficient for SR edge estimation | 0.01 |
| $edge\_prior$ | Default value for edges' SR | 0.5 |
| $shortest_paths$ | Number of shortest paths to sample from | 5 |
| $shortest\_safest\_ratio$ | Ratio of alternation between shortest and safest paths | 0.5 |

