# OpenReview forum: "Help Me Explore: Combining Autotelic and Social Learning via Active Goal Queries"
_ICLR.cc/2023/Conference — Submitted to ICLR 2023_

### Official Review · Reviewer_sim2 · 2022-10-24

**Confidence:** 4
**Correctness:** 3
**Technical Novelty And Significance:** 3
**Empirical Novelty And Significance:** 3
**Recommendation:** 5

**Clarity, Quality, Novelty And Reproducibility:**

The paper and overarching motivation is clear. Components of this work exist in prior work, but the underlying algorithm and block manipulation environment seem novel. Code is provided for reproducibility.

**Strength And Weaknesses:**

Strengths
- The underlying intuition of querying a ‘social partner’ for goal suggestions just beyond the agent’s capabilities makes sense, and learning when to query for that information with a mostly autotelic agent is an interesting problem.
- The chosen task seems difficult and interesting, and the authors show ablations across various settings of their method to justify the mixture of social and autotelic learning.

Weaknesses
- It’s not clear how well this method extends beyond the block stacking environment, as (from my understanding) the social partner has to be hard-coded with underlying knowledge of the task to create a zone of proximal development which is used to select the ‘beyond’ goals. This creates a bit of an unfair comparison with the baseline ACL methods, which propose goals without any underlying expert knowledge of the task / difficulty progression.
- Not clear what authors mean by ‘current ACL strategies are all limited to generate goals from distribution of effects agent has already experienced’ — eg. methods like GoalGAN (Florensa et al., 2018) try to generate goals of intermediate difficulty, which may not be goals the agent has seen/achieved before.
- Furthermore, in some tasks it may not be straightforward to create a graph showing progression of goal difficulties, as with block stacking — e.g. locomotion across changing/mixed terrain. Rather than directly giving expert knowledge, perhaps another way of showing the usefulness of social learning without necessarily using expert knowledge is to query a human teacher instead, or to simulate imperfect social partners by injecting some noise into the proposed ‘beyond’ goals?

Additional Feedback
- Figure 1 could be made larger, currently the text is hard to read. Same with Figure 2.
- Fourth line under 3.2 typo: goal-conditioned ‘lagent’

**Summary Of The Paper:**

The authors propose an exploration method inspired by the importance of social learning in humans. The method ‘Help Me Explore’ combines individual epodes for practicing already known goals with active learning via ‘social queries’ for goals just beyond the agent’s current abilities, as suggested by a social partner. Specifically, social queries are made when the learning progress on known goals is low. The primary task the paper focuses on is a 5 block rearrangement task.

**Summary Of The Review:**

I recommend a 5. While the idea is interesting, the paper would be strengthened with a wider set of experimental environments (where it would be harder to hard-code a social partner) and/or querying humans as social partners. Currently the comparison against ACL methods seem a little unfair, as those methods do not make use of expert knowledge of what the zone of proximal development would be, but HME does, with the hard-coded social partner.

---

> ### Author Response · Authors · 2022-11-15
> **Response to Reviewer sim2**
>
> As reviewer qrSi, the reviewer is questioning the generality of our work because the social partner (SP) is hard-coded and environment specific. Again, as we stressed in our general response, in principle we would expect a human to fill this role and we do not consider the SP as part of our method. Our point is more than calling upon a knowledgeable SP helps circumvent the limits of goal exploration based on intrinsic motivations based on mutual information or any other information-theoretic criterion.
>
> We agree that including a knowledgeable SP makes the comparison with Automatic Curriculum Learning (ACL) methods which do not benefit from such an SP unfair, but our point is precisely that the ACL methods all come with inherent limits that our approach is meant to circumvent via social help.
>
> So we agree with the reviewer that the next step of our work will consist in making our agent robust to mistakes from the SP, as a human SP cannot be expected to always behave optimally.

---

### Official Review · Reviewer_qrSi · 2022-11-02

**Confidence:** 5
**Correctness:** 3
**Technical Novelty And Significance:** 3
**Empirical Novelty And Significance:** 3
**Recommendation:** 5

**Clarity, Quality, Novelty And Reproducibility:**

The paper is well-written, the message is clear. The figures are decent, and convey a lot of information. The novelty is decent, the specific form of the shaped goal curriculum is new. The reproducibility is poor: the appendix is clearly unfinished, many hyper-parameters are missing, a large part of the agent design is not discussed, the authors don’t discuss compute hardware, or how they tuned hyper-parameters (and which ones).

**Strength And Weaknesses:**

Strengths
- The problem is well-framed and motivated
- The empirical validation is thorough and the results are backing up the proposed methods
- The head-line result of stacking 5 blocks is compelling
- The links to social learning are interesting and likely to inspire follow-up work

Weaknesses
- The specific “social partner” is broadly named, but quite tailored to the specific stacking task: how general is the approach if there is a less obvious decomposition into sequential stepping stones (e.g. how could this system teach an agent how to play StarCraft?)
- I think the paper should discuss its limitations more explicitly: e.g. how important is the full observability assumption made in Sec 3.1?
- There is a lack of clarity on some important aspects, e.g. around episode boundaries, termination, whether reward is accumulated after a goal is achieved, whether each episode is reserved for a single goal, etc.
- The reliable success on 5-block stacking is great: why not try 6 blocks to see where/whether the method breaks?
- The hand-crafted evaluation set is a bit disappointing, would there not be a way to define criteria for “interesting, non-overlapping, complex goals” and have the set emerge from data? That would boost generality of the methodology?
- The on-demand SP querying section seems very ad-hoc: why does Eq (1) contain an exponential if its argument never strays far from 0 (i.e it is essentially linear)? How does the LP difference take into account the fact that the early and later goal sets may differ? How important/justified is the indirection to compress success rates into a neural network instead of just using counts to determine LP? (and how much does its generalization rely on the goal representation?) Also, how do you prevent biased predictions in that network, caused by under-training on rare goals?

Smaller concerns
- It might be nice to tie back the conclusion to the first sentence of the intro: what type of open-ended learning is feasible now that wasn’t before?
- Don’t introduce the identical mathcal{C} and mathcal{G} under two notations, pick one and clarify that paragraph
- The last paragraph of Sec 3.1 is both vague (exploration, discovery, mastery are all undefined) and incorrect: you don’t seek the “maximum number of goals”, but rather mastery of a specific set of hard-to-reach ones.
- Fig 1: how do discovered goals inform the social partner (top arrow)?
- End of Sec 1: mention this is a simulated robotics environment, not an actual robotic arm
- The references are quite heavily relying on a particular Bordeaux research group, you might want to re-balance those a little.
- Sec 3.1: I’d recommend using indicator function notation for the reward, instead of the pythonic “==”.
- Is the query probability re-assessed “each new episode” as stated in Sec 3.3, or held constant actress chunks of 6000 episodes?
- Sec 4.1: how do you sample 20 goals from the C1 set that just contains 10 (Table 1)?
- VDS: how do the 3 nets preserve diversity? What’s the softmax temperature? Why once do max-over-50 and once softmax-over-1000?
- Sec 4.2 “indeed [not] good stepping stones”: can you show this?
- Can you add a plot of P_query over the course of training?
- Fig 3 could benefit from log-scale, and definitely is confusing to have one out of 8 subplot use a different y-scale to the rest…
- Table 2 would probably be clearer as an xy-plot?
- Table 2: what is “w/o intern”? The results you would have gotten without the help of an intern :-)
- Please use 7% instead of 6.98% – given the large error-bars, they are equally correct, but one of them is readable too.
- Sec 5: 0.5% is 1250 episodes, not 12500?
- There are multiple duplicate citations
- Appendix A1: either refer to what it adds to the paper from the main text, or omit the section
- Table 4: what are “shortest and safest paths”? Is is a table that belongs into another paper?

Typos
- “lagents” (twice)
- numerous missing commas, namely in front of “which” clauses
- “leftest” -> “left-most”
- discoverS a wider
- Vygotsky’S concept




**Summary Of The Paper:**

This paper introduces a goal-shaping method to overcome sparse exploration challenges. The key idea is to provide a goal-conditioned learning agent with both “frontier” and “beyond” goals; but only on demand, when unaided learning stagnates. Empirical demonstrations show this can achieve stacks of 5 blocks in a simulated robotics task.

**Summary Of The Review:**

This is a well-written paper with an interesting new idea, backed up by solid empirical evidence. It has a number of major flaws, and lots of minor ones, however, and I really hope the post-rebuttal version will address them, so that I can advocate for its acceptance with good conscience.

--- update post rebuttal ---
Unfortunately, the authors did not address my questions in any satisfying way, nor update the submission.

---

> ### Author Response · Authors · 2022-11-15
> **Response to Reviewer qrSi**
>
> We warmly thank the reviewer for their numerous constructive comments and suggestions, and for their positive assessment of our work.
>
> As we stressed in our general response, the social partner is a more-knowledgeable agent that will guide the exploration of the learner. As long as it is hard-coded, it will need to be tailored to the specific task but, eventually, when sample efficiency will let humans fill that role, there will be no task-specific knowledge incorporated in the design. In other words, the social partner is not part of our proposed algorithm but more of an environment module. In StarCraft, a social partner could propose to first go to places, build N fighters, defend place X, etc. In that case, the space of possible goals is wider and less well-defined but there are still stepping stone skills to help develop more sophisticated ones; a curriculum that a social partner might help uncover.
>
> We like very much the idea of pushing the limits of our system towards dealing with 6, 7 or even more blocks, though the computational cost grows much more than linearly in the number of blocks and we are concerned about the carbon footprint of our work.
>
> We do not provide a specific answer to each item in the reviewer’s list of comments, suggestions and typos, but again we are very grateful about them and will use them to build a better version of our work.

---

> > ### Comment · Reviewer_qrSi · 2022-11-24
> > **missed opportunity**
> >
> > The rebuttal phase is a great opportunity for papers like this that hover near the threshold: a revised submission can be significantly improved, and engaging with reviewers can lead to a dialogue that brings out the best in a paper.
> >
> > In this case, the authors missed the opportunity. I'm surprised they did not opt for submitting a revised version, and I did expect more answers: My review contained literally 24 questions, only 1 was answered.
> > I adjusted my overall recommendation (downward).

---

### Official Review · Reviewer_JZLF · 2022-11-03

**Confidence:** 4
**Clarity, Quality, Novelty And Reproducibility:** The quality and novelty are questiona…
**Correctness:** 2
**Technical Novelty And Significance:** 1
**Empirical Novelty And Significance:** 1
**Recommendation:** 3

**Strength And Weaknesses:**

Weaknesses:
1. The paper starts with grandiose claims of tackling "open-ended learning". However, open-ended learning involves learning to perform across diverse environments. But the definition of open-ended learning in this work seems restricted only to learning different skills in a given environment. In experiments, it is mostly restricted to goal conditioned environments and learning a goal conditioned policy.
2. "But where do goals come from? Almost always, they are sampled from a fixed distribution over a predefined goal space; i.e. they come from an engineer." There are numerous works where the goals are NOT generated from a fixed distribution (listed in the references below)
3. The previous statement makes us believe that in this work, the goals are not generated from a fixed distribution. However, a few paragraphs later, the authors note that "In this second challenge — the one we focus on — agents must learn to organize their own learning trajectories by prioritizing goals with the objective of maximizing long-term skill mastery." i.e, this work focuses on learning a goal conditioned policy from pre-defined goals.
4. "In social episodes, a social partner suggests a novel goal to the agent and decomposes it into two consecutive sub-goals: 1) a frontier goal that the agent already discovered and, if it is reached, 2) a beyond goal never achieved by the agent but just beyond the its current abilities." The social agent keeps a list of all the goals discovered so far and a list of all the goals to be reached. This is not tractable in most environments.
5. One of the contributions listed is: "an active learning mechanism allowing the agent to self-monitor its learning progress and, when it stagnates, query the social partner for a goal suggestion". This seems like a standard active learning setting and not a novel contribution.

References:
[1] Learning with AMIGo: Adversarially Motivated Intrinsic Goals. Campero et al, 2020
[2] Intrinsic Motivation and Automatic Curricula via Asymmetric Self-Play. Sukhbaatar et al, 2017
[3] Asymmetric self-play for automatic goal discovery in robotic manipulation. OpenAI et al, 2021
[4] An automatic curriculum for learning goal-reaching tasks. Zhang et al, 2021
[5] Automatic curriculum learning through value disagreement. Zhang et al, 2020
[6] Exploration via hindsight goal generation. Ren et al, 2019
[7]  Automatic goal generation for reinforcement learning agents. Florensa et al, 2018


**Summary Of The Paper:**

This paper proposes to learn a goal conditioned policy based on goals generated from a pre-defined goals and the agent's own goals.


**Summary Of The Review:**

Based on the weaknesses listed, I recommend to reject this paper

---

> ### Author Response · Authors · 2022-11-15
> **Response to Reviewer JZLF**
>
> We thank the reviewer for rightfully questioning our positioning and providing a set of useful references.
>
> The reviewer mainly challenges our claim that we are addressing an open-ended learning problem. Though a widely adopted definition of what open-ended learning truly means is still missing (see e.g. discrepancies between  [1] and [2]), we have to admit the reviewer is right. In the next version of our paper, we will rather claim that our HME approach addresses one of the important subproblems of the open-ended learning challenge: finding an efficient curriculum in a very large goal space with many unreachable goals, and where some goals are much harder than others to achieve.
>
> [1] Doncieux, S., Filliat, D., Díaz-Rodríguez, N., Hospedales, T., Duro, R., Coninx, A., ... & Sigaud, O. (2018). Open-ended learning: a conceptual framework based on representational redescription. Frontiers in neurorobotics, 12, 59.
>
> [2] Stooke, A., Mahajan, A., Barros, C., Deck, C., Bauer, J., ... & Czarnecki, W. M. (2021). Open-ended learning leads to generally capable agents. arXiv preprint arXiv:2107.12808.

---

### Official Review · Reviewer_LbgX · 2022-11-03

**Confidence:** 4
**Correctness:** 1
**Technical Novelty And Significance:** 3
**Empirical Novelty And Significance:** 1
**Recommendation:** 1

**Clarity, Quality, Novelty And Reproducibility:**

Issues in clarity discussed above:

* minor typographical issues
* claims extend beyond evidence
* ordering of ideas makes it challenging to comprehend
* axes on plots are left unlabeled, making it difficult to interpret empirical results
* terms are used before being introduced

**Strength And Weaknesses:**

Strengths:

1. This paper takes care to embed its understanding social agents within the broader interdisciplinary literature.

Concerns:

1. **In figure 2a, the success rate of multiple agents is depicted across episodes. There are a few aspects of this plot that I find concerning**

i. I am concerned that the experiment was run over hours rather than a standardised number of time-steps or episodes. By limiting the learning by wall-time, both the algorithm and the implementation of an agent are being evaluated. This means that code-level implementation choices that are independent of the learning method being evaluated will influence the reported performance. This is not a fair comparison.

ii. I am concerned that the results are reported over 5 seeds. Reporting the standard deviation over results for so few seeds is misleading. Moreover, given the results are overlapping in many locations, it's difficult for me to conclude that HME outperforms the comparators.

iii. I am struggling to understand the star notation in figure 2a. In this case, I'm not sure what the "left-most" algorithm is. I am assuming it is HME-50. It's unusual to report statistical differences across time-series data as presented. It would be better to report over summary statistics (error after learning). I have doubts that with 5 seeds that the data meets the requirements for a T-test (e.g., normality). The paper should provide information that demonstrates that this data meets the assumptions for the tests that have been done.

you're more likely to find a statistically significant difference when you're making many comparisons. To account for this, you should use a Bonferroni correction (or similar). In that case, it would require dividing by the number of comparisons made. As a result alpha will be less than 0.05, I believe. This would mean that the evaluation episodes for which there is a statistically significant difference would decrease.

Reporting the actual P values would be helpful in this case.

iv. At this point, I might wonder why the confidence intervals were not plotted, given the comparisons made. If you have the confidence intervals, why would standard deviation be reported for the error bars?

v. It is claimed that "non-social baselines reach a plateau of around 0.75". Examining the figure, it seems that all but go-exp ss achieve scores higher than 0.75, and that the average final performance of most are somewhere around 0.8-0.9. This looks like a misrepresentation of the data.

vi. It looks like HME-50 and HME-50b perform equitably in terms of success rate. This suggests that there is little difference in final performance between suggesting goals that are exclusively beyond the agent's perceived skill level, and suggesting goals that are at the frontier of the agent's skill.

2. **some aspects of the evaluation strategy are unclear**

The agent is evaluated on "interesting evaluation goals that humans could easily describe". What makes a goal interesting or easy to describe isn't elaborated on, making it challenging to assess whether this is a reasonable evaluation strategy.

3. **Figure 2b could be interpreted in several ways**

I am struggling to understand what is being reporting in figure 2b. To my understanding, It counts the number of positions that are reached by the agent that are---according to the SP agent---adjacent to unknown configurations (configurations that have yet to be achieved?). In this case, HME-50 B reaches more stepping stone configurations across most episodes with lower variance(?) than HME-50. If I understand this correctly, it means that the agent is reaching more positions that are perceived by the SP to be at the frontier of the underlying agent's ability.

it is concluded that *"this suggests that the role of the frontier goal is to enable the agent to reach the
beyond goal from the first time."* If I understand this claim correctly, we cannot conclude this from the data presented. All we can say is that the agent that is proposing goals beyond its skill level is able to reach the frontier more regularly.

A plot that depicts the number of new goals an agent reaches, or the rate at which the frontier expands, would be better able to support the claims being made in the paper, to my understanding.

i. Figure 3 is mentioned twice in the text, from what I can tell. There is no Y axis labeled, and there is no description in the caption: I cannot tell what is being plotted here without hunting through the text. In the text an average success metric, and a social request metric are defined. I don't know what it means for the agents to have a score of 600 in one of the categories. I do not know what the error bars are.

Labelling is absent from figure 5 as well.

ii. The axes are different for each sub-plot, making them difficult to visually compare.

iii.  "As shown in the top row of Figure 3, ACL agents do not even discover the hardest configurations." In the top row of figure 3, I see HME-50 included. It would be helpful if you direct the reader's attention to specific sub-figures.

4. **It is unclear why 50 was the beta parameter chosen for the baseline comparisons.**
Examining figure 5  seems to suggest that HME-20 has the best performance (although there is higher variance).

5. (stylistic suggestion) Figure 4 is challenging to interpret. In this figure, the success rate for all beta values chosen is presented. In this case, I cannot readily see the difference between the beta values---especially for the less complex stacking classes---because all of the lines overlap. Grouping the lines by category and presenting all the beta values on a single plot would help with the interpretability. Right now, it is challenging for me to see the difference between the success rates. For instance, one plot with HME 20, 50, 100, 200 plotted for the success rate of S4.

6. **Frequently the claims of the paper extend beyond the truth.**

In a couple of locations *very* strong claims are made.

* *agents must learn to organize their own learning trajectories by prioritizing goals
with the objective of maximizing long-term skill mastery*

*Must* is a very strong word. Agents may benefit from prioritising goals, but it's not a necessity. Certainly many sparse reward problems have been solved by other means.

* *[e-greedy action selection] is not enough to experience informative rewards in hard exploration problems*

If an agent is following a random behaviour policy, it can experience reward in sparse environments (as defined in the paper). It might be highly improbable, but it is certainly not impossible.

Minor points:

1. Terms are used without being defined (e.g., SP is used several times before being defined in 3.2)
2. Some of the citations feel a bit strange. For instance, Mnih et al. 2015 is used as a citation for epsilon-greedy action selection.




**Summary Of The Paper:**

This paper argues that artificial agents may benefit from socially-guided skill discovery. This paper presents a protocol for active learning that enables an agent to query a partner for a goal. The partner is an auxiliary agent that maintains a model of the probability that an agent could reach goals in the goal-space. The paper claims that this agent outperforms existing baselines, including curriculum-based agents.

**Summary Of The Review:**

* In several places, the claims of the paper extend beyond the evidence
* plots are not clearly explained, making it a challenge to interpret the empirical claims

For these reasons, I suggest a strong reject.

---

> ### Author Response · Authors · 2022-11-15
> **Response to Reviewer LbgX (1/2)**
>
> We thank the reviewer for their very precise troubleshooting comments on our work which will help us improve it. The main concern of the reviewer is about the statistical significance and validity of our results. We answer the reviewer’s concerns using the numbering they provided in their review.
>
> 1.i. By contrast with what the reviewer understood, the agents benefit from a fixed budget in terms of episodes, not in wall clock time (see e.g. the x-axis in Fig 2a).
>
> 1.ii. Figure 2a showcases baselines and ablations of the proposed algorithm (HME-50). This figure answers, for each variant X: does the mean performance of HME-50 significantly outperform the mean performance of X? Regularly across training episodes, this statistical test is applied and the stars depicted indicate significant differences with respect to our algorithm HME-50, the color of the star indicating which of the variants is tested against HME-50. We agree that we need to clarify this in the legend. Note that standard deviations can cross and the difference might still be significant. Standard deviations indicate the breadth of the distribution, while standard errors of the mean would evaluate the error on the mean estimate.
>
> 1.iii. This evaluation protocol is detailed and justified in [1]. 5 seeds is indeed a low number. It is however comparable to 90% of RL papers nowadays, because of the high computing costs required to run such algorithms. This said, the statistical tests take the number of seeds into account: with lower sample sizes, the test simply requires stronger evidence (larger difference in the samples’ means) to reject the null hypothesis. The reviewer is right to question the validity of the assumptions on the normality of the distribution. Indeed, 5 seeds are not enough to demonstrate a normal distribution. However, the empirical studies conducted in [1] show that the Welch’s t-test is overall robust to violations of this assumption for a variety of distributions (bi-modal, asymmetric) and respect the false positivity rate (it does not attribute significant differences more than it should, i.e. more than the significance level alpha). With few seeds, this test is the best we can use.
>
> The correction for multiple comparison aspects is also discussed in Colas et al (2019). The reviewer would be right if the criterion for overall success was “there is at least one significant difference across learning”. In that case, multiple comparisons would scale the chances of this happening by chance and would require Bonferroni corrections. Here, most comparisons towards the end of learning are significant (maybe 80-90%). If we were to assume that 5% of them were obtained by chance (our significance level is alpha=5), the rest of the differences would still be significant. In short, this evaluation protocol across learning brings more evidence to the reader for them to make up their mind compared to the traditional p-value at the end of learning, although, of course, 5% of these differences might not be significant.
>
> 1.iv. Despite the excellent work of [2] – that we discovered too late to adopt for this paper –, almost everyone in the field presents standard deviations and they let us appreciate the diversity of performance one might end up with. Confidence interval would instead estimate possible errors in the estimate of the mean performance. This is a different kind of information and we cannot report both on a single graph. The stars give some information about errors in the mean estimates since they evaluate whether the difference between two means is statistically significant.
>
> 1.v. The reviewer is right, we will improve the visualization of experimental results in the next version of the paper.
>
> 1.vi. Yes, again the reviewer is right, we need to better investigate this point to determine whether suggesting goals beyond the frontier truly makes a difference w.r.t. suggesting goals at the frontier and letting the agent explore from there.
>
> 2. The list of evaluation categories and evaluation goals are precisely defined in the paper (Table 1 and Appendix Table 3). We picked them to be diverse, to encompass configurations of various difficulties and to include the most difficult configurations, i.e. the farthest from the initial co-planar configuration. Our world defines a very large goal space where the exhaustive set of achievable goals is unknown. This is even more the case in open-ended worlds where the goal space is unbounded. In such worlds, there is no objective set of evaluation goals, evaluation must be subjective. Defining measurable criteria would not help as we would not be able to crawl the whole goal space to detect all the goals satisfying these criteria. The best we can do is to report an evaluation set that contains a diversity of goals we, as human evaluators, consider interesting and that span various levels of complexity such that we can measure forms of progress.

---

> > ### Author Response · Authors · 2022-11-15
> > **Response to Reviewer LbgX (2/2)**
> >
> >
> > 3. We agree that Figure 2b needs to be reworked and we thank the reviewer for their suggestions on how to better stress our point.
> >
> > 3.i. The axis of Figures 3 and 5 are defined in the figures’ captions but should also be defined on the graphs. This will be corrected.
> >
> > 3.ii and iii. We agree that we need to rework on these figures to better stress our points. The next version of the paper will take this into account.
> >
> > 4. The reviewer is right, we chose HME-50 because of the lower variance, but we might have chosen HME-20.
> >
> > 5. We thank the reviewer for the suggestion.  We will follow the suggestion in the next version of the paper.
> >
> > 6. The reviewer is right that some of our claims need to be rephrased. We will do so in the next version of the paper.
> >
> > [1] Colas, C., Sigaud, O., & Oudeyer, P. Y. (2019). A hitchhiker's guide to statistical comparisons of reinforcement learning algorithms. arXiv preprint arXiv:1904.06979.
> >
> > [2] Agarwal, R., Schwarzer, M., Castro, P. S., Courville, A. C., & Bellemare, M. (2021). Deep reinforcement learning at the edge of the statistical precipice. Advances in neural information processing systems, 34, 29304-29320.

---

### Author Response · Authors · 2022-11-15
**General response to all reviewers**

We thank the reviewers for their time and effort in providing useful feedback about our paper. A key point that we have poorly highlighted in our paper is that we do not consider the social partner (SP) in this work as part of our algorithm. Our perspective in future work is to use a human as SP, but we need to robustify the HME approach before we can do this in practice. As a consequence of this perspective, we are not much concerned if we need to endow the SP with some ad-hoc, environment specific knowledge. We believe this is why, despite showing results on another case study in the appendix, we have failed to convince the reviewers about the generality and significance of our approach and we need another writing strategy to better highlight our point, which is that where autonomous exploration encounters some inherent limits (see e.g. [1]), having a social partner dragging the learner beyond these limits is an efficient option.

Below we briefly answer the reviewers’ individual points.

[1]: Campos, V., Trott, A., Xiong, C., Socher, R., Giró-i-Nieto, X., & Torres, J. (2020, November). Explore, discover and learn: Unsupervised discovery of state-covering skills. In International Conference on Machine Learning (pp. 1317-1327). PMLR.

---

### Decision · Program_Chairs · 2023-01-20

**Decision:**

Reject

**Justification For Why Not Higher Score:**

After reading the author response, the reviewers were in consensus to reject the paper.

**Justification For Why Not Lower Score:**

N/A

**Metareview: Summary, Strengths And Weaknesses:**

Summary:
This paper proposes a method of socially-provided goal-shaping for a learning algorithm, to provide assistance when individual learning stagnates.  The proposed method is shown to be effective in enabling a learning agent to accomplish a 5-block stacking task when individual learning does not succeed in learning the task.

Strengths:
The reviewers liked the broad interdisciplinary scope of the paper, and the potential research direction around social learning.  They also liked the motivation and the block stacking results.

Weaknesses:
The reviewers found many weaknesses.  There was a lack of understanding on the how general the method is within or across domains (reviewer qrSi, sim2). There is a lack of understanding on the interaction between the goal space and the social partner (reviewer JZLF). There was also concern on the experiment design and evaluation (reviewer LbgX) - though the authors addressed several of these immediate concerns, and promised more in a future paper iteration.  The paper overclaims in several instances (reviewer LbgX), and the authors acknowledged that noted comments were overclaims and need to be addressed in the next version of the paper.